# Clinical Outcomes, Patient-Reported Outcomes, and Economic Burden for Thai People Living with Chronic Urticaria (CORE-CU) in routine practice: A study protocol for a monocentric prospective longitudinal study

**Mati Chuamanochan**[1,2]☯*, **Surapon Nochaiwong**[2,3]☯*

**1** Division of Dermatology, Department of Internal Medicine, Faculty of Medicine, Chiang Mai University, Chiang Mai, Thailand, **2** Pharmacoepidemiology and Statistics Research Center (PESRC), Faculty of Pharmacy, Chiang Mai University, Chiang Mai, Thailand, **3** Department of Pharmaceutical Care, Faculty of Pharmacy, Chiang Mai University, Chiang Mai, Thailand

☯ These authors contributed equally to this work.
* mati.c@cmu.ac.th (MC); surapon.nochaiwong@gmail.com (SN)

**Data Availability Statement:** No datasets were generated or analysed during the current study. All

## Abstract

### Background

Few prospective longitudinal studies have been conducted in Thailand to account for the long-term response to chronic urticaria (CU) treatment, clinical outcomes, and patient-reported outcomes (PROs) among people living with CU based on routine practice. As such, a prospective longitudinal study will be conducted to better understand the long-term responses to treatment options and the burden of disease in Thai CU patients.

### Methods and design

This study is a routine clinical practice registry-based, monocentric, prospective, observational longitudinal study in the northern region of Thailand. Adult patients in an outpatient clinic diagnosed with CU, including both chronic spontaneous urticaria and chronic inducible urticaria will be recruited for this study. The cohort will be collected and registered using the joint routine clinical practice data based on multiple datasets including claims outpatient and inpatient data, routine laboratory results, medication utilization, health care costs, clinical characteristics, long-term urticaria care and monitoring, and PRO measures. The point prevalence of adverse health outcomes will be estimated and reported corresponding to 95% confidence intervals (95% CIs). The overall trend analysis will be analyzed to explore the effect of over time across the cohort time frame.

### Conclusion

This prospective longitudinal study will report the clinical outcomes, PROs, and economic burden among Thai people living with CU based on routine clinical practice. Findings will

relevant data from this study will be made available upon study completion.

**Funding:** Pharmacoepidemiology and Statistics Research Center (PESRC) through the Chiang Mai University provides to SN The funders had and will not have a role in study design, data collection and analysis, decision to publish, or preparation of the manuscript.

**Competing interests:** The authors have declared that no competing interests exist.

provide comprehensive evidence and could facilitate best practices for CU care management for health care professionals, researchers, policymakers, and public society.

## Trial registration

Thai Clinical Trials Registry (TCTR, thaiclinicaltrials.org) registration TCTR20210706005. Registered on July 6, 2021.

## Introduction

Chronic urticaria (CU) is a chronic inflammatory skin disease characterized by the presence of recurrent hives (wheals), and/or angioedema for longer than six weeks [1]. According to global epidemiology estimates, CU is a common disease that affects adults, with a point prevalence of 0.86% [2]. However, it has been revealed to increase over time, in particular, up to 1.5% in Latin America and 1.4% in Asian countries [2]. CU is recognized as chronic spontaneous urticaria (CSU, lesions occur spontaneously), chronic inducible urticaria (CIndU, lesions occur in response to explicit triggers), or both (co-existence of CSU and CIndU). With respect to the subtypes of CU, approximately two-thirds of patients with CU were characterized as CSU [3], and 20.2% of patients were diagnosed as having a combination of CSU and CIndU [4].

On the basis of patient and society's perspective, individuals living with CU are significantly negatively affected by a wide spectrum of disease burdens, including high impact on health-related quality of life (HRQOL), impaired school/work performance, limited activity and physical function, worsened mental health and well-being issues, psychosocial problems (i.e., impaired family life, partnerships, and social interactions), and high costs and economic burden [5–9]. Moreover, existing studies have revealed that CU patients place a significant burden of illness higher than other chronic skin diseases [9, 10].

Although the current European Academy of Allergology and Clinical Immunology (EAACI)/Global Asthma and Allergy European Network (GA$^2$LEN)/European Dermatology Forum (EDF; EuroGuiDerm)/Asia Pacific Association of Allergy, Asthma and Clinical Immunology (APAAACI) international guidelines suggest effective licensed doses of non-sedating second-generation H1-antihistamines as the first-line therapy for the management of CU and there is substantial evidence of its effectiveness [1, 11], CU patients are often inadequately controlled for urticaria symptoms with this first-line therapy [3, 8, 12, 13].

To the best of the researcher's knowledge, there are very few prospective longitudinal studies in Thailand to account for the long-term response of CU treatment, clinical outcomes, and patient-reported outcomes (PROs) among people living with CU based on routine practice. As such, this study aim to develop the CORE-CU (Clinical Outcomes, Patient-Reported Outcomes, and Economic Burden for Thai People Living with Chronic Urticaria), a monocentric prospective cohort study, to better understand the long-term responses to treatment options and the disease burden in Thai CU patients.

## Objectives

### Primary objectives.

i. To evaluate and track the treatment response and clinical outcomes among adult CU patients during routine urticaria care management.

ii. To assess the impact of CU on long-term PROs, including HRQOL, symptom burden, mental health issues, well-being, and psychosocial problems.

**Secondary objectives.**

i. To establish and validate the clinical utility of the PROs in daily practice.

ii. To explore the economic burden of people living with CU, including medication utilization, health care costs, and out-of-pocket costs, in the Thai CU cohort.

# Material and methods

## Study design

The Clinical Outcomes, Patient-Reported Outcomes, and Economic Burden for Thai People Living with Chronic Urticaria (CORE-CU) is a routine clinical practice registry-based, monocentric, prospective, observational longitudinal study at the urticaria clinic, Maharaj Nakorn Chiang Mai Hospital, a northern region of Thailand. This study was registered with the Thai Clinical Trials Registry (thaiclinicaltrials.org; TCTR20210706005, July 6, 2021).

## Study population

Adult patients in an outpatient urticaria clinic will be consecutively recruited for this research. The study population will consist of adult patients diagnosed with CU (duration >6 weeks) based on the EAACI/GA$^2$LEN/EuroGuiDerm/APAAACI Guideline on urticaria [1]. Both CSU and CIndU are included (Table 1). The inclusion and exclusion criteria for the longitudinal cohort are shown in Table 2.

## Data collection and study schedule

The cohort will be collected and registered using the routine clinical practice data of adult patients with CU. Data will be embedded based on multiple datasets, including (i) electronic health records, outpatient and inpatient claims data, and routine laboratory results; (ii) the Support System Pharmacy Dispensing extract, an administrative data source on pharmacy dispensing that provides patient-level details on medication utilization, dispensing detail, and

**Table 1. Classification of chronic urticaria subtypes [1].**

| |
|---|
| **Chronic spontaneous urticaria (CSU)** |
| • Spontaneous appearance of wheals, angioedema or both due to known (i.e, auto-reactivity—that is the presence of the mast-cell activating auto-antibodies) or unknown causes |
| **Chronic inducible urticaria (CIndU)** |
| • Symptomatic dermographism (urticaria factitia or dermographic urticaria) |
| • Delayed pressure urticaria (pressure urticaria) |
| • Solar urticaria |
| • Aquagenic urticaria |
| • Contact urticaria |
| • Cold urticaria (cold contact urticaria) |
| • Heat urticaria (heat contact urticaria) |
| • Cholinergic urticaria |
| • Vibratory angioedema |

**Table 2. Eligibility criteria of the CORE-CU study.**

| Inclusion criteria | Exclusion criteria |
|---|---|
| • Participants aged ≥18 years at the date of screening | • Any conditions (both mental or physical) that would interfere with the participant's ability to comply with the study protocol |
| • Diagnosed with chronic urticaria based on the EAACI/GA²LEN/EDF/APAAACI guideline: had wheals and/or angioedema for >6 weeks | • The prognosis for survival <12 months |
| • Had an ability to understand and willingness to sign an informed consent statement | • A primary skin disorder that could explain the pruritus or angioedema (i.e., cholestatic liver disease, atopic dermatitis, bradykinin-mediated angioedema, Bullous pemphigoid, Well's syndrome, Schnitzler's syndrome, etc.) |

Abbreviations: APAAACI, Asia Pacific Association of Allergy, Asthma and Clinical Immunology; CORE-CU, Clinical Outcomes, patient-Reported outcomes, and Economic burden for Thai people living with Chronic Urticaria; EAACI, European Academy of Allergology and Clinical Immunology; EDF, European Dermatology Forum (EuroGuiDerm); GA²LEN, Global Asthma and Allergy European Network; HIV/AIDS, human immunodeficiency virus/acquired immune deficiency syndrome.

health care costs; and (iii) Urticaria Care Database via REDCap$^{TM}$, which provides patient-level details on sociodemographic and clinical characteristics, long-term urticaria care and monitoring, and PRO measures.

Participant recruitment will be performed continuously during the observation period to identify patients who are interested in participating in the CORE-CU registered cohort. Eligible participants will be monitored based on routine practice every 3 months (medical history, physical examination, and health care costs; further laboratory tests during study follow-up are needed in the case of clinically indicated) based on in-center follow-up care. Regarding PROs assessment, urticaria symptoms responses using participant self-reported diaries tools will be assessed monthly-basis. While HRQOL and mental health issue aspects will be evaluated and collected every 3 months during the CU clinic follow-up care. All the participants will be monitored for as long as possible; however, eligible participants will be assessed for at least one year (Table 3). If possible, after completing a one-year follow-up, participants will be reassessed and monitored based on the schedule of observation and procedures in the first year manner. Ultimately, participants will be followed and censored until the date of the earliest incidence of death, withdrawal for any reasons, transfer to another center, or loss to follow-up; whichever occurred first.

To minimize participant dropout and missing data over time, the communications team and case manager staff members will facilitate cohort retention by employing various reminder approaches. Eligible participants will be contacted one day before the date of in-center visiting or the date of anticipated data collection based on participants' preference contact methods (text message, phone, social media, or direct mail). During routine in-center follow-up care (every 3 months), participants will receive personalized health feedback and CU care management. Moreover, the study team will also share a lay language summary of study findings when available.

Participants will be asked to complete the urticaria symptoms and control diaries based on the international EAACI/GA²LEN/EuroGuiDerm/APAAACI guideline for the management of CU using daily urticaria activity score (daily UAS) and urticaria activity score over 7 days (UAS7). PRO measures, and health care utilization and costs were collected at baseline along with routine clinical urticaria monitoring visits.

Participant characteristics and routine laboratory data included the following:

**Table 3. Schedule of observation and procedures.**

| Parameter | Screening -1 | Assessment and Follow-Up | | | | | | | | | | | | |
| --- | --- | --- | --- | --- | --- | --- | --- | --- | --- | --- | --- | --- | --- | --- |
| | | 0 | 1 | 2 | 3 | 4 | 5 | 6 | 7 | 8 | 9 | 10 | 11 | 12 |
| Check eligibility against inclusion/exclusion criteria and medication review | X | | | | | | | | | | | | | |
| Gathering informed consent | | X | | | | | | | | | | | | |
| Sociodemographic and lifestyle data | | X | | | | | | | | | | | | |
| Medical history and physical examination | | X | | | X | | | X | | | X | | | X |
| Patient-reported outcomes/experiences | | | | | | | | | | | | | | |
| • Once daily/weekly UAS | | X | X | X | X | X | X | X | X | X | X | X | X | X |
| • Short Form UCT | | X | X | X | X | X | X | X | X | X | X | X | X | X |
| • PatGA-VAS | | X | X | X | X | X | X | X | X | X | X | X | X | X |
| • PatGA-LS | | X | X | X | X | X | X | X | X | X | X | X | X | X |
| • CU-Q$_2$oL | | X | | | X | | | X | | | X | | | X |
| •  • DLQI | | X | | | X | | | X | | | X | | | X |
| • EQ-5D-5L | | X | | | X | | | X | | | X | | | X |
| • WHO-5 Well-being Index | | X | | | X | | | X | | | X | | | X |
| • PHQ-9 | | X | | | X | | | X | | | X | | | X |
| • GAD-7 | | X | | | X | | | X | | | X | | | X |
| • SSS-8 | | X | | | X | | | X | | | X | | | X |
| • ISI | | X | | | X | | | X | | | X | | | X |
| • WPAI: GH V2.1 | | X | | | X | | | X | | | X | | | X |
| • MAST | | X | | | X | | | X | | | X | | | X |
| • Satisfaction with care | | X | | | X | | | X | | | X | | | X |
| Routine laboratory tests (performed locally) | | X | | | | | | | | | | | | |
| Record medication changes | | X | | | X | | | X | | | X | | | X |
| Health care costs: direct medical cost, direct non-medical cost, and indirect cost | | X | | | X | | | X | | | X | | | X |
| Data monitoring | | X | | | X | | | X | | | X | | | X |
| Statistical analysis and reporting | | | | | | | | | | | | | | X |

Abbreviations: CU-Q$_2$oL, Chronic Urticaria Quality of Life Questionnaire; DLQI, Dermatology life Quality Index; GAD-7, Generalized Anxiety Disorder 7-item; ISI, Insomnia Severity Index; MAST, Medication Adherence Scale in Thais; PatGA-LS, Patients's Global Assessment of Disease Control-Likert Scale; PatGA-VAS, Patient's Global Assessment of Disease Severity-Visual Analogue Scale; PHQ-9, Patient Health Questionnaire 9-item; SSS-8, Somatic Symptom Scale 8-item; UAS, Urticaria Activity Score; UCT, Urticaria Control test; WPAI, Work Productivity and Activity Impairment Instrument.

- Sociodemographic data (i.e., age, sex, marital status, education, income, health insurance, smoking and alcohol status, and body mass index)

- Medical history (i.e., history of drug/food/substance allergy, comorbid conditions, subtypes of CU) and physical examination

- Routine and urticaria-related laboratory tests (i.e., complete blood count, coagulogram, bio-chemistry, liver function test, thyroid function and anti-thyroid autoantibodies test, erythrocyte sedimentation rate, C-reactive protein, d-dimer, fibrinogen, total immunoglobulin E, antinuclear antibody test, serum levels of complement components [C3 and C4] and its activity [CH50], hepatitis B surface antigen, anti-hepatitis C virus antibody, stool parasite test, and *Helicobacter pylori* infection [if indicated])

- Medication utilization and adherence

## Pre-specified outcomes of interest

The pre-specified outcomes of interest and assessment of the CORE-CU cohort included the following (Table 3):

i. Clinical outcomes

- Urticaria symptoms and severity using the daily urticaria activity score (daily UAS) and urticaria activity score over 7 days (UAS7) [1]

- Urticaria symptom control using the Short Form Urticaria Control Test (UCT) [14], Patient's Global Assessment of Disease Control-Likert Scale (PatGA-LS) [15], and Patient's Global Assessment of Disease Severity-Visual Analog Scale (PatGA-VAS) [15]

- Safety profiles: occurrence of adverse events

ii. PRO measures

- HRQOL using the Chronic Urticaria Quality of Life Questionnaire (CU-Q2oL) [16], Dermatology Life Quality Index (DLQI) [17], and EuroQoL-5 dimension-5 level (EQ-5D-5L)

- Mental health issues and psychosocial problems using the Patient Health Questionnaire 9-item (PHQ-9) [18], Generalized Anxiety Disorder 7-item (GAD-7) [19], and the 5-item World Health Organization Well-being Index (WHO-5) [20]

- Symptom burden and impact on sleep using the Somatic Symptom Scale 8-item (SSS-8) [21] and Insomnia Severity Index (ISI) [22]

- Medication adherence using the Medication Adherence Scale in Thais (MAST) [23]

- Satisfaction with care using the global visual analog scale-10 cm and our team development version tool

- Work impairment using the Work Productivity and Activity Impairment Instrument (WPAI) [24]

iii. Health care utilization and economic outcomes

- Health care costs: direct medical costs, direct non-medical costs, and indirect costs

- Hospitalization, emergency visits, and extra/unplanned visits during the follow-up

## Statistical analysis plan

**Sample size estimation.** The sample size of participants was estimated based on the prevalence of adverse outcomes among CU patients, including (i) inadequate response to a standard dose of H1-antihistamines or uncontrolled urticaria symptoms (UCT score <12) with a range of 28.7%–82.8% [3, 8, 12, 13]; (ii) HRQOL impairment (DLQI >10 points) with a range of 30.9%– 31.5% [7, 8]; and (iii) mental health and psychosocial problems (depression, anxiety, insomnia, and somatic symptoms) with a range of 17.2%–55.7% [6, 7, 9]. With respect to a design effect of 2.0 and missing data of 20%, a target of at least 132 CU patients will be required for this study to ensure a 10% of margin of error and 0.05 type I error. Nevertheless, there will be no restrictions on the number of participants in this prospective longitudinal study.

**Statistical analysis.** Preplanned statistical analysis will be performed based on one-year follow-up care (Table 3). The characteristics of the study cohort will be described as frequencies with percentages for categorical variables and mean ± standard deviation or medians with interquartile range for continuous variables as suitable. The differences between treatment

groups (treatment-based non-sedating second-generation H1-antihistamines, biologics, or immunosuppressive agents) or severity strata of urticaria symptoms based on UAS7 (mild, 7–15 points; moderate, 16–27 points; and severe, 28–42 points) for categorical and continuous variables will be compared using Fisher's exact test and analysis of covariance or Kruskal-Wallis test, respectively.

The point prevalence of adverse health outcomes (e.g., treatment responses [response vs. refractory to first-line therapy with non-sedating second-generation H1-antihistamines], symptoms controlled [uncontrolled, UCT <12 points vs. controlled, UCT ≥12 points], and occurrence of adverse mental health issues [including HRQOL impairment, depression, anxiety, insomnia, and somatic symptoms) will be estimated and reported corresponding to 95% confidence intervals (95% CIs). Subgroup analyses will be performed based on key patient characteristics (i.e., age of study participants, female sex, duration of CU, severity of urticaria symptoms, subtypes of CU, body weight/body mass index, total immunoglobulin E level, history of allergy or other systemic diseases, specific pharmacological treatments, and concomitant and rescue medications). In addition, one-year follow-up findings will be compared to assess the long-term response to the prevalence of adverse health outcomes. The overall trend analysis will explore the effect of over time across the cohort time frame.

For cross-sectional analyses, the crude association among participant characteristics will be explored using the univariable logistic regression models to identify the candidate risk factors for adverse health outcomes. Subsequently, candidate risk factors with a $P$-value of <0.100 will be included in the multivariable logistic regression analysis with the backward elimination method. The variance inflation factors of risk factors within the multivariable model will be investigated to detect the multicollinearity of the final model. The odds ratios (ORs) with 95% CIs will be reported as the effect estimates of the final set of risk factor models for adverse health outcomes. For longitudinal analyses, changes in the adverse health outcome status during the follow-up period will be analyzed using multilevel logistic regression models with random intercepts to explore the associations with individual-level and baseline predictors.

All analyses will be performed using Stata software (version 16.0; StataCorp LP). Statistical significance for all tests will be two-tailed, with a $P$-value of <0.05. For participants with missing covariates, multiple imputation methods will be used to analyze incomplete datasets. However, variables containing more than 20% of missing data will be excluded from all analyses.

**Data management and security.** During the study follow-up, paper-based information and participants' self-reported PROs will be continuously entered into the Urticaria Care Database via the REDCap$^{TM}$ platform using a double data entry and double-check approach. Two trained health information professionals from the Pharmacoepidemiology and Statistics Research Center (PESRC) will extract and obtain the predefined data from the multiple datasets as mentioned (electronic health records, Support System Pharmacy Dispensing extract, and Urticaria Care Database).

To ensure accurate data assessment and limit missing data, an external panel of health information professionals from the PESRC—well trained in urticaria care management will independently cross-check, verify, and validate the datasets for the high-quality data collection system. All information will be organized and encrypted using a password. Only the management team can access all data. Moreover, the completeness of the case record form and other essential documents will be checked and monitored throughout the progress of the study on time.

## Ethical consideration

**Ethics approval and consent to participate.** This study was approved by the Institutional Review Board of the Faculty of Medicine, Chiang Mai University (MED-2564-08319). The

CORE-CU study will be performed in accordance with the 1964 Declaration of Helsinki and amendments or comparable ethical standards. All protocol amendments will be submitted to the Ethics Committee for approval. Study protocol version 2, dated August 6, 2021, was used to assemble this manuscript.

**Informed consent, autonomy, and confidentiality.** Written informed consent will be obtained from all participants before they are included in the study. The participants' consent will be obtained by trained research assistants. Based on the International Conference on Harmonization: Thai Guideline for Good Clinical Practice, a thumbprint is obligatory when a participant is illiterate. All participants have the right to leave a specific question unanswered or withdraw informed consent at any time for any reason during the study period. Data collected prior to study withdrawal will be considered for participants who discontinue. A written lay statement will be provided to inform them that the decision not to participate in the study will not affect following clinical care at the study site. No gifts or other payments will be offered.

Data gathered are stored in the study database without identifiers, and linking information is kept separate in a locked filing cabinet. The unique identifiers will be generated, completely encrypted, coded, and will be recorded as double-entry for use, principally for statistical analysis purposes using computer software. Additionally, the collected data will remain confidential and shared only with authorized research members.

**Role of funder.** The funder of the study had no role in the study design, collection, analysis, interpretation of the data, or writing of the report. The author had full access to all the data in the study and had final responsibility for the decision to submit it for publication.

## Patient/Public involvement and dissemination of results

**Patient and public involvement.** The participants and the public had no role in the trial design, recruitment, conduct, and monitoring. However, written lay summary results will be available to patients and the public.

**Dissemination of results.** Findings from this study will be disseminated through scientific meetings and publications in peer-reviewed journals. The researchers will report the results in accordance with the Reporting of studies Conducted using Observational Routinely-collected health Data (RECORD) Statement [25] along with the Guidelines for Inclusion of Patient-Reported Outcomes in Clinical Trial Protocols (SPIRIT-PRO Extension) [26]. Any modification will be described succinctly in the final report.

## Discussion

CU is a common chronic inflammatory skin disease with a substantial impact on a patients' daily life and productivity loss (impaired school/work performance) [5]. At present, treatment options and pharmacological therapies, including H1-antihistamines, anti-inflammatory, immunomodulatory, immunosuppressant, and biological agents, are available for the management of CU [11, 27–29]. Unfortunately, people living with CU are often resistant to standard treatment with the licensed dose of H1-antihistamines [3, 8, 12, 13].

From the patient and their family's perspectives, living with CU significantly reduced HRQOL, limited social interactions, and had an economic impact. Subsequently, negative emotions and adverse mental health outcomes may occur owing to a high long-term burden, as the nature of the disease is unpredictable and debilitating with no curative treatment [6, 7, 9]. From the providers' perspective, CU also places significant health care utilization including treatment medications, laboratory tests, outpatient visits, extra visits and emergency department visits, and hospitalizations.

Real-world evidence with respect to treatment practices and long-term clinical outcomes of CU in Thailand is limited. Moreover, very little is known about the burden of disease on PROs and its economic impact on Thai patients. Taken together, comprehensive real-world evidence is needed to support the best practices for CU management care. To the best of our knowledge, this is the first routine practice-based prospective longitudinal study among Thai people living with CU. This study would document comprehensive long-term outcomes and the impact of disease on patients' daily lives. Given the monocentric health care system under the universal coverage scheme in Thailand, the generalizability of our findings to other CU centers may be limited.

## Cohort status

The CORE-CU cohort is currently in the participant enrollment phase. To date, a total of 75 eligible patients have participated as of August 10, 2022.

## Conclusions

This prospective longitudinal study will report the clinical outcomes, PROs, and economic burden among Thai people living with CU based on routine clinical practice. From the clinician's and patient's perspectives, these findings have the potential to provide comprehensive evidence and facilitate best practices for CU care management for health care professionals, researchers, policymakers, and public society. The results from this study will be disseminated through scientific meetings and publications in peer-reviewed journals.

## Acknowledgments

We thank all study participants, research assistants, and administrative staff involved in the study. Particular thanks to the following individuals for their contribution to the project: Arun Kunti, Pharmacoepidemiology and Statistics Research Center (PESRC), Faculty of Pharmacy, Chiang Mai University; and Angkana Kittichaiyakorn, Chronic Urticaria Clinic, Maharaj Nakorn Chiang Mai Hospital.

## Author Contributions

**Conceptualization:** Mati Chuamanochan, Surapon Nochaiwong.

**Data curation:** Mati Chuamanochan, Surapon Nochaiwong.

**Formal analysis:** Surapon Nochaiwong.

**Funding acquisition:** Mati Chuamanochan, Surapon Nochaiwong.

**Investigation:** Mati Chuamanochan.

**Methodology:** Mati Chuamanochan, Surapon Nochaiwong.

**Supervision:** Mati Chuamanochan, Surapon Nochaiwong.

**Writing – original draft:** Mati Chuamanochan, Surapon Nochaiwong.

**Writing – review & editing:** Mati Chuamanochan, Surapon Nochaiwong.

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
