## [Decision Letter · Decision Letter 0]

28 Jun 2022

PONE-D-21-27924

Clinical Outcomes, Patient-Reported Outcomes, and Economic Burden for Thai People Living with Chronic Urticaria (CORE-CU) in Routine Practice: A Study Protocol for a Monocentric Prospective Longitudinal Study

PLOS ONE

Dear Dr. Chuamanochan,

Thank you for submitting your manuscript to PLOS ONE. After careful consideration, we feel that it has merit but does not fully meet PLOS ONE’s publication criteria as it currently stands. Therefore, we invite you to submit a revised version of the manuscript that addresses the points raised during the review process.

I would like to sincerely apologize for the delay you have incurred with your submission. It has been exceptionally difficult to secure reviewers to evaluate your study. We have now received two completed reviews; the comments are available below. The reviewers have raised significant scientific concerns about the study that need to be addressed in a revision.

Please revise the manuscript to address all the reviewer's comments in a point-by-point response in order to ensure it is meeting the journal's publication criteria. Please note that the revised manuscript will need to undergo further review, we thus cannot at this point anticipate the outcome of the evaluation process.

We look forward to receiving your revised manuscript.

Kind regards,

Miquel Vall-llosera Camps

Senior Editor

PLOS ONE

Journal Requirements:

Additional Editor Comments:

Please clarify the status of the participant recruitment: has yet to begin, its ongoing or has already finished?

Reviewers' comments:

Reviewer's Responses to Questions

**Comments to the Author**

1. Does the manuscript provide a valid rationale for the proposed study, with clearly identified and justified research questions?

Reviewer #1: Yes

Reviewer #2: Yes

2. Is the protocol technically sound and planned in a manner that will lead to a meaningful outcome and allow testing the stated hypotheses?

Reviewer #1: Yes

Reviewer #2: Partly

3. Is the methodology feasible and described in sufficient detail to allow the work to be replicable?

Reviewer #1: Yes

Reviewer #2: No

4. Have the authors described where all data underlying the findings will be made available when the study is complete?

Reviewer #1: Yes

Reviewer #2: No

5. Is the manuscript presented in an intelligible fashion and written in standard English?

Reviewer #1: Yes

Reviewer #2: Yes

6. Review Comments to the Author

You may also provide optional suggestions and comments to authors that they might find helpful in planning their study.

Reviewer #1: Thank you for your work on this manuscript and the opportunity to review it. On the whole, I considered your presentation organized and insightful. I would like to offer a few thoughts about the manuscript overall. The first item is to be careful about making the tense be unified throughout. There were several points in the manuscript when I felt confused if you were reporting something that has occurred or something that will occur over the course of the study. Additionally, I thought there could have been slightly more detail on the timeframe associated with the data collection and the outcomes to be reported. As someone unfamiliar with chronic urticaria, I am uncertain what routine clinical care and schedules resemble and I wished I had some additional clarity around this point in thinking about your research study.

Narrowing my comments to a few specific items in your manuscript, there are several points where additional clarification might be beneficial for the reader.

1. Study population, 3rd line: The second sentence ends in the word “urticarial.” I couldn’t tell if this was an error or if there was a word missing (or perhaps this is correct).

2. Data collection, study schedule and pre-specified outcomes of interest, paragraph 1, line 1: What does the word “joint” mean in this context? If this is intended to reflect the major data sources that are enumerated after this sentence, I believe this word can be removed for clarity.

3. Same paragraph, line 4: “...an administrative data” should be “an administrative data set” or “an administrative data source.”

4. Same paragraph: Could you provide a little more detail about the “Urticaria Care Database?” It is uncertain if this is something unique to your facility or if it is a broader resource.

5. Same section, paragraph 2: The second sentence indicates that participants will complete the “urticaria symptoms and control diaries.” Can you provide additional detail about what these are? Perhaps a bit more information about the frequency of the data collection and for how long it will be collected? Also, is this something that requires specific procedures for analysis (I am uncertain if these will yield narrative or qualitative data)?

6. Same paragraph: This has a very long sentence at the end which enumerates a number of baseline and control variables. Please consider reformatting this in some way (perhaps a bulleted list or table?) to make this easier to digest.

7. Same section, description of outcomes: As mentioned above, I felt that there could be a bit more description about the timeframe for the outcomes you plan to report. Table 3 seems to indicate that several of the outcomes will be collected monthly and others quarterly. It is the reader’s assumption that the specific time point of interest is for reporting 12-month outcomes.

8. Do participants return to the clinic once per month? Will data collection be altered or be conducted differently if the patient does not return to the clinic? Will 12-month outcomes be reportable for a participant if they provide data at month 11 but not at month 12? Is there a window around the 12-month time point that represents an acceptable range for which you will report an outcome as being collected at 1 year?

9. Statistical analysis, paragraph 1, sentence 2: You note how “differences between treatment groups or severity strata” will be handled. Can you explain more about what the treatment groups are and what the severity strata are?

10. Same section, next paragraph: Are the adverse health outcomes that you plan to analyze defined or can they be defined a priori? If so, it would seem appropriate to do so. In the event that the definitions are the same as those you described in the sample size estimation section, please note this somehow.

11. Same paragraph, last sentence: I believe you meant to use “over time” instead of “overtime.”

12. Informed consent, autonomy and confidentiality, paragraph 2: “All data were anonymous and were not useful in identifying participants.” This sentence is unclear to me. Is your intention to indicate that the screening process for determining potentially eligible patients did not include patient identifiers?

13. Discussion, paragraph 3: “This study would provide comprehensive long-term outcomes and the impact of disease on patients’ daily lives.” Instead of “provide,” would “evaluate” or “document” be a more appropriate word here?

14. Conclusions, sentence 2: Similar to the point above, since the study is not yet completed, I would suggest rewording this sentence to note that the findings either “have the potential to provide” or “seek to provide” comprehensive evidence.

15. Table 3: “Gartering informed consent” likely should be “garnering” or “gathering” instead.

Thank you again for your work on this manuscript. Overall, you present a good plan for your research and wish you success in executing the study.

Reviewer #2: The authors have provided a strong rationale for this study and the findings will be an important contribution to the literature and may improve clinical care for Thai people with chronic uticaria.

Here are questions and comments for the authors to consider:

- I am unclear about the current status of recruitment into the study. The manuscript suggests that the participants have already been recruited ("...signed consent"), but goes on to describe how the participants will be prospectively recruited. If recruitment has already commenced, the authors should comment on how actual enrollment has gone relative to projected enrollment, etc.

The following questions relate to data collection and management:

- The authors say that participants will be followed for as long as possible and at least for one year. In the schedule of observations and procedures, it appears as though there are routine clinical follow-ups at 3, 6, 9, and 12 months and that the clinical data collection will coincide with these time points. The PROs appear to be collected at these time points as well, but also during the interim months. The authors should clarify the official study visit schedule. The authors should also clarify what it means to be followed "for as long as possible". Will follow-ups that extend beyond the 12 month period occur at the same frequency (i.e., every 3 months)? Will the data collection continue to include clinical and PROs?

- What will be considered the study database? Is it the REDCap database? How will the data be entered into the database? At one point, it appears as though data collection will be on paper forms (there is a statement that the forms will be kept in locked storage).

- The authors should describe how each type of data will be collected and entered into the database. Are the Research Assistants that are helping obtain consent going to do data collection? How will the clinical data from the medical record system and the pharmacologic system be abstracted and entered into the study database?

- How will the PRO data be collected? Will the patients be asked to complete forms during their visits at 3, 6, 9, and 12 months? How will the PRO data be collected during the interim periods, e.g., 4 months, 5 months, and so on?

- The depression instrument- the PHQ9- has a suicidality item. How will the research team monitor for endorsement of this item and what actions will be taken to address the participant's mental health needs? If there are insufficient resources to be responsive to this item, the research team could consider using the PHQ8 which drops the suicide item.

- Will concerted efforts be made to retain participants in the study? Will research team members remain in contact with participants to remind them to complete their study visits, etc? If participants don't show up for visits, will they be contacted and will there be attempts to reschedule them?

- How will the above efforts work in conjunction with the data review and validation processes? Will the panel generate reports or review reports that are generated by someone else? Who will maintain the REDCap database? How will have access to the database? It may be helpful for the panel to include a 3rd member in case there is a need to adjudicate particular outcomes.

- What will monitoring efforts look like for data collection that extends beyond the 12 month period (i.e., once the cohort all completes the 12 month follow-up)?

7. PLOS authors have the option to publish the peer review history of their article (what does this mean?). If published, this will include your full peer review and any attached files.

Reviewer #1: No

Reviewer #2: **Yes: **Lauren Allen, DrPH

---

## [Author Response · Author response to Decision Letter 0]

1 Oct 2022

Response to Reviewers

PONE-D-21-27924 entitled, “Clinical Outcomes, Patient-Reported Outcomes, and Economic Burden for Thai People Living with Chronic Urticaria (CORE-CU) in Routine Practice: A Study Protocol for a Monocentric Prospective Longitudinal Study”

Comments to the Author

Reviewer 1

Thank you for your work on this manuscript and the opportunity to review it. On the whole, I considered your presentation organized and insightful. I would like to offer a few thoughts about the manuscript overall. The first item is to be careful about making the tense be unified throughout. There were several points in the manuscript when I felt confused if you were reporting something that has occurred or something that will occur over the course of the study. Additionally, I thought there could have been slightly more detail on the timeframe associated with the data collection and the outcomes to be reported. As someone unfamiliar with chronic urticaria, I am uncertain what routine clinical care and schedules resemble and I wished I had some additional clarity around this point in thinking about your research study.

Narrowing my comments to a few specific items in your manuscript, there are several points where additional clarification might be beneficial for the reader.

Thank you very much for your insightful comments. We have addressed all issues that reviewers mentioned as follows.

#1. Study population, 3rd line: The second sentence ends in the word “urticarial.” I couldn’t tell if this was an error or if there was a word missing (or perhaps this is correct).

Thank you for your pertinent observation. We found some error in reporting the final version of the word “urticarial”. By checking, we have made corrections and changed to “Urticaria” throughout the manuscript.

#2. Data collection, study schedule and pre-specified outcomes of interest, paragraph 1, line 1: What does the word “joint” mean in this context? If this is intended to reflect the major data sources that are enumerated after this sentence, I believe this word can be removed for clarity.

Thank you very much for your suggestion. It has been revised and made changes as recommended. Please read as: “The cohort will be collected and registered using the routine clinical practice data of adult patients with CU”.

#3. Same paragraph, line 4: “...an administrative data” should be “an administrative data set” or “an administrative data source.”

We have made changes as recommended. Please read as: “the Support System Pharmacy Dispensing extract, an administrative data source on pharmacy dispensing that provides patient-level details on medication utilization, dispensing detail, and health care costs”.

#4. Same paragraph: Could you provide a little more detail about the “Urticaria Care Database?” It is uncertain if this is something unique to your facility or if it is a broader resource.

Thank you for your insightful comment. We have addressed this issue with respect to the “Urticaria Care Database” as read “Urticaria Care Database via REDCapTM, which provides patient-level details on sociodemographic and clinical characteristics, long-term urticaria care and monitoring, and PRO measures”.

#5. Same section, paragraph 2: The second sentence indicates that participants will complete the “urticaria symptoms and control diaries.” Can you provide additional detail about what these are? Perhaps a bit more information about the frequency of the data collection and for how long it will be collected? Also, is this something that requires specific procedures for analysis (I am uncertain if these will yield narrative or qualitative data)?

Thank you for your suggestions. We have added information on the “urticaria symptoms and control diaries as follows:

“Participants will be asked to complete the urticaria symptoms and control diaries based on the international EAACI/GA2LEN/EuroGuiDerm/APAAACI guideline for the management of CU using daily urticaria activity score (daily UAS) and urticaria activity score over 7 days (UAS7).”

#6. Same paragraph: This has a very long sentence at the end which enumerates a number of baseline and control variables. Please consider reformatting this in some way (perhaps a bulleted list or table?) to make this easier to digest.

Thank you for your suggestions. We have reorganized as recommended under the “Data collection and study schedule” section using a bullet. 

#7. Same section, description of outcomes: As mentioned above, I felt that there could be a bit more description about the timeframe for the outcomes you plan to report. Table 3 seems to indicate that several of the outcomes will be collected monthly and others quarterly. It is the reader’s assumption that the specific time point of interest is for reporting 12-month outcomes.

Based on the routine practice care and open dynamic cohort design for data collection, meaning that participants can leave or be added over time. Participants are continually added when they are diagnosed with CU and monitored (including extra follow-up visit due to worsening or uncontrolled symptoms), so roughly they will be monitored based on routine practice (every 3 months). In contrast with the fixed cohort and monitoring, participants can also leave the cohort owing to free of urticaria symptoms, which reflects the real situation of chronic urticaria disease. As such, we have provided an anticipated time frame under the “Data collection and study schedule” section and a specific schedule for monitoring in Table 3 as follows:

“Participant recruitment will be performed continuously during the observation period to identify patients who are interested in participating in the CORE-CU registered cohort. Eligible participants will be monitored based on routine practice every 3 months (medical history, physical examination, and health care costs; further laboratory tests during study follow-up are needed in the case of clinically indicated) based on in-center follow-up care. Regarding PROs assessment, urticaria symptoms responses using participant self-reported diaries tools will be assessed monthly-basis. While HRQOL and mental health issue aspects will be evaluated and collected every 3 months during the CU clinic follow-up care. All the participants will be monitored for as long as possible; however, eligible participants will be assessed for at least one year (Table 3). If possible, after completing a one-year follow-up, participants will be reassessed and monitored based on the schedule of observation and procedures in the first year manner. Ultimately, participants will be followed and censored until the date of the earliest incidence of death, withdrawal for any reasons, transfer to another center, or loss to follow-up; whichever occurred first.”

#8. Do participants return to the clinic once per month? Will data collection be altered or be conducted differently if the patient does not return to the clinic? Will 12-month outcomes be reportable for a participant if they provide data at month 11 but not at month 12? Is there a window around the 12-month time point that represents an acceptable range for which you will report an outcome as being collected at 1 year?

Thank you very much for your concerns. With respect to the routine practice care and open dynamic cohort design for data collection, all available data will be reported using the last observation forward approached. However, all analyses will be addressed for missing data as mentioned in the “Statistical analysis” section as follows:

“For participants with missing covariates, multiple imputation methods will be used to analyze incomplete datasets. However, variables containing more than 20% of missing data will be excluded from all analyses.”

However, we try to minimize participant dropout and missing data using various reminder approaches as mentioned under the “Data collection and study schedule” section: 

“To minimize participant dropout and missing data over time, the communications team and case manager staff members will facilitate cohort retention by employing various reminder approaches. Eligible participants will be contacted one day before the date of in-center visiting or the date of anticipated data collection based on participants’ preference contact methods (text message, phone, social media, or direct mail).”

#9. Statistical analysis, paragraph 1, sentence 2: You note how “differences between treatment groups or severity strata” will be handled. Can you explain more about what the treatment groups are and what the severity strata are?

Thank you very much for your concerns. We have addressed this issue as follows:

“The differences between treatment groups (treatment-based non-sedating second-generation H1-antihistamines, biologics, or immunosuppressive agents) or severity strata of urticaria symptoms based on UAS7 (mild, 7-15 points; moderate, 16-27 points; and severe, 28-42 points) for categorical and continuous variables will be compared using Fisher’s exact test and analysis of covariance or Kruskal-Wallis test, respectively.”

#10. Same section, next paragraph: Are the adverse health outcomes that you plan to analyze defined or can they be defined a priori? If so, it would seem appropriate to do so. In the event that the definitions are the same as those you described in the sample size estimation section, please note this somehow.

We have added more information regarding this issue as read: “The point prevalence of adverse health outcomes (e.g., treatment responses [response vs. refractory to first-line therapy with non-sedating second-generation H1-antihistamines], symptoms controlled [uncontrolled, UCT <12 points vs. controlled, UCT ≥12 points], and occurrence of adverse mental health issues [including HRQOL impairment, depression, anxiety, insomnia, and somatic symptoms) will be estimated and reported corresponding to 95% confidence intervals (95% CIs)”. 

#11. Same paragraph, last sentence: I believe you meant to use “over time” instead of “overtime.”

Thank you for your suggestion, we have changed as recommended as read: “The overall trend analysis will explore the effect of over time across the cohort time frame”.

#12. Informed consent, autonomy and confidentiality, paragraph 2: “All data were anonymous and were not useful in identifying participants.” This sentence is unclear to me. Is your intention to indicate that the screening process for determining potentially eligible patients did not include patient identifiers?

To make it more clear, we have rephrased this sentence as read: “All data were anonymous and are not able to identify the participants’ information”.

#13. Discussion, paragraph 3: “This study would provide comprehensive long-term outcomes and the impact of disease on patients’ daily lives.” Instead of “provide,” would “evaluate” or “document” be a more appropriate word here?

Thank you for your suggestion. We have made changes accordingly as read: “This study would document comprehensive long-term outcomes and the impact of disease on patients’ daily lives”.

#14. Conclusions, sentence 2: Similar to the point above, since the study is not yet completed, I would suggest rewording this sentence to note that the findings either “have the potential to provide” or “seek to provide” comprehensive evidence.

Thank you so much for your point. We totally agree and it has been changed as recommended accordingly as read: “From the clinician’s and patient’s perspectives, these findings have the potential to provide comprehensive evidence and facilitate best practices for CU care management for health care professionals, researchers, policymakers, and public society”. 

#15. Table 3: “Gartering informed consent” likely should be “garnering” or “gathering” instead.

Table 3 has been corrected for an error as recommended.

Thank you again for your work on this manuscript. Overall, you present a good plan for your research and wish you success in executing the study.

Kindly thanks again for your valuable comments.

Reviewer 2

The authors have provided a strong rationale for this study and the findings will be an important contribution to the literature and may improve clinical care for Thai people with chronic uticaria.

Here are questions and comments for the authors to consider:

#1. I am unclear about the current status of recruitment into the study. The manuscript suggests that the participants have already been recruited ("...signed consent"), but goes on to describe how the participants will be prospectively recruited. If recruitment has already commenced, the authors should comment on how actual enrollment has gone relative to projected enrollment, etc.

Thank you for your concerns. Based on the routine practice care and open dynamic cohort design for data collection, meaning that participants are continually added when they are diagnosed with CU and monitored. We anticipated to recruit participants at least of 132 CU patients to ensure a 10% of margin of error and 0.05 type I error. Nevertheless, there will be no restrictions on the number of participants in this prospective longitudinal study. Moreover, the current status of the study has been addressed under the “Cohort status” section as follows: 

“The CORE-CU cohort is currently in the participant enrollment phase. To date, a total of 75 eligible patients have participated as of August 10, 2022.”

The following questions relate to data collection and management:

#2. The authors say that participants will be followed for as long as possible and at least for one year. In the schedule of observations and procedures, it appears as though there are routine clinical follow-ups at 3, 6, 9, and 12 months and that the clinical data collection will coincide with these time points. The PROs appear to be collected at these time points as well, but also during the interim months. The authors should clarify the official study visit schedule. The authors should also clarify what it means to be followed "for as long as possible". Will follow-ups that extend beyond the 12 month period occur at the same frequency (i.e., every 3 months)? Will the data collection continue to include clinical and PROs?

Thank you for your pertinent observation. Based on the routine practice care and open dynamic cohort design for data collection, meaning that participants can leave or be added over time. Participants are continually added when they are diagnosed with CU and monitored, so roughly they will be monitored based on routine practice (every 3 months). To make it more clear, we have stated this issue under the “Data collection and study schedule” section.

“Participant recruitment will be performed continuously during the observation period to identify patients who are interested in participating in the CORE-CU registered cohort. Eligible participants will be monitored based on routine practice every 3 months (medical history, physical examination, and health care costs; further laboratory tests during study follow-up are needed in the case of clinically indicated) based on in-center follow-up care. Regarding PROs assessment, urticaria symptoms responses using participant self-reported diaries tools will be assessed monthly-basis. While HRQOL and mental health issue aspects will be evaluated and collected every 3 months during the CU clinic follow-up care. All the participants will be monitored for as long as possible; however, eligible participants will be assessed for at least one year (Table 3). If possible, after completing a one-year follow-up, participants will be reassessed and monitored based on the schedule of observation and procedures in the first year manner. Ultimately, participants will be followed and censored until the date of the earliest incidence of death, withdrawal for any reasons, transfer to another center, or loss to follow-up; whichever occurred first.”

#3. What will be considered the study database? Is it the REDCap database? How will the data be entered into the database? At one point, it appears as though data collection will be on paper forms (there is a statement that the forms will be kept in locked storage).

Specifically, REDCapTM is a mature, secure web application for building and managing online surveys and databases. While REDCapTM can be used to collect virtually any type of data in any environment, it is specifically geared to support online and offline data capture for research studies and operations. In this case, apart from the electronic health records and an administrative data source on pharmacy dispensing, participant-level details on sociodemographic, self-reporting PROs, as well as urticaria symptoms diaries throughout study follow-up using paper forms will be entered into the REDCapTM platform. 

Thus, our cohort will be collected and registered using the routine clinical practice data of adult patients with CU as mentioned in the “Data collection and study schedule” and “Data management and security” sections. 

“Data will be embedded based on multiple datasets, including (i) electronic health records, outpatient and inpatient claims data, and routine laboratory results; (ii) the Support System Pharmacy Dispensing extract, an administrative data source on pharmacy dispensing that provides patient-level details on medication utilization, dispensing detail, and health care costs; and (iii) Urticaria Care Database via REDCapTM, which provides patient-level details on sociodemographic and clinical characteristics, long-term urticaria care and monitoring, and PRO measures.”

AND

“During the study follow-up, paper-based information and participants’ self-reported PROs will be continuously entered into the Urticaria Care Database via the REDCapTM platform using a double data entry and double-check approach. Two trained health information professionals from the Pharmacoepidemiology and Statistics Research Center (PESRC) will extract and obtain the predefined data from the multiple datasets as mentioned (electronic health records, Support System Pharmacy Dispensing extract, and Urticaria Care Database).”

#4. The authors should describe how each type of data will be collected and entered into the database. Are the Research Assistants that are helping obtain consent going to do data collection? How will the clinical data from the medical record system and the pharmacologic system be abstracted and entered into the study database?

Details of data collection are stated under the “Data collection and study schedule” section, including participant characteristics and routine laboratory data, and pre-specified outcomes of interest. Moreover, we have added information regarding this issues under the “Data management and security” section as follows:

“During the study follow-up, a paper-based information and participants self-reported PROs will be continuously entered into the Urticaria Care Database via REDCapTM platform using a double data entry and double check approach. Two trained health information professionals from the Pharmacoepidemiology and Statistics Research Center (PESRC) will extract and obtain the predefined data from the multiple datasets as mentioned (electronic health records, Support System Pharmacy Dispensing extract, and Urticaria Care Database).”

AND

“To ensure accurate data assessment and limit missing data, an external panel of health information professionals from the PESRC—well trained in urticaria care management will independently cross-check, verify, and validate the datasets for the high-quality data collection system. All information will be organized and encrypted using a password. Only the management team can access all data. Moreover, the completeness of the case record form and other essential documents will be checked and monitored throughout the progress of the study on time.”

#5. How will the PRO data be collected? Will the patients be asked to complete forms during their visits at 3, 6, 9, and 12 months? How will the PRO data be collected during the interim periods, e.g., 4 months, 5 months, and so on?

Details regarding PROs data collected are described under the “Data collection and study schedule” as read: “Regarding PROs assessment, urticaria symptoms responses using participant self-reported diaries tools will be assessed monthly-basis. While HRQOL and mental health issue aspects will be evaluated and collected every 3 months during the CU clinic follow-up care”.

#6. The depression instrument- the PHQ9- has a suicidality item. How will the research team monitor for endorsement of this item and what actions will be taken to address the participant's mental health needs? If there are insufficient resources to be responsive to this item, the research team could consider using the PHQ8 which drops the suicide item.

Thank you for your concerns. Based on our daily practice, all mental health assessment tools, including depression, anxiety, as well as suicide ideation were used as routine screening tools rather than diagnosed tools. Once, adverse mental issues were identified, a collaborative team including psychiatrist and psychologists will be consulted for further evaluation and treatment care in a timely manner. As such, we decided to keep all mental health assessment tools in the study based on our routine protocol practice care. 

#7. Will concerted efforts be made to retain participants in the study? Will research team members remain in contact with participants to remind them to complete their study visits, etc? If participants don't show up for visits, will they be contacted and will there be attempts to reschedule them?

Thank you for your insightful comments. Information regarding strategies for increasing study retention has been described under the “Data collection and study schedule” section as read: 

“To minimize participant dropout and missing data over time, the communications team and case manager staff members will facilitate cohort retention by employing various reminder approaches. Eligible participants will be contacted one day before the date of in-center visiting or the date of anticipated data collection based on participants’ preference contact methods (text message, phone, social media, or direct mail). During routine in-center follow-up care (every 3 months), participants will receive personalized health feedback and CU care management. Moreover, the study team will also share a lay language summary of study findings when available.”

#8. How will the above efforts work in conjunction with the data review and validation processes? Will the panel generate reports or review reports that are generated by someone else? Who will maintain the REDCap database? How will have access to the database? It may be helpful for the panel to include a 3rd member in case there is a need to adjudicate particular outcomes.

Thank you for your concerns. The REDCapTM platform was maintained by the Research Administration System, Faculty of Medicine, Chiang Mai University (https://redcap.med.cmu.ac.th/). As mentioned, data validation and monitoring has been stated under the “Data management and security” section as read:

“To ensure accurate data assessment and limit missing data, an external panel of health information professionals from the PESRC—well trained in urticaria care management will independently cross-check, verify, and validate the datasets for the high-quality data collection system. All information will be organized and encrypted using a password. Only the management team can access all data. Moreover, the completeness of the case record form and other essential documents will be checked and monitored throughout the progress of the study on time.”

#9. What will monitoring efforts look like for data collection that extends beyond the 12 month period (i.e., once the cohort all completes the 12 month follow-up)?

Thank you for pertinent observation. To address this issues, we have added more information under the “Data collection and study schedule” section as read: 

“All the participants will be monitored for as long as possible; however, eligible participants will be assessed for at least one year (Table 3). If possible, after completing a one-year follow-up, participants will be reassessed and monitored based on the schedule of observation and procedures in the first year manner. Ultimately, participants will be followed and censored until the date of the earliest incidence of death, withdrawal for any reasons, transfer to another center, or loss to follow-up; whichever occurred first.”

---

## [Decision Letter · Decision Letter 1]

21 Nov 2022

PONE-D-21-27924R1Clinical Outcomes, Patient-Reported Outcomes, and Economic Burden for Thai People Living with Chronic Urticaria (CORE-CU) in Routine Practice: A Study Protocol for a Monocentric Prospective Longitudinal StudyPLOS ONE

Dear Dr. Chuamanochan,

Thank you for submitting your manuscript to PLOS ONE. After careful consideration, we feel that it has merit but does not fully meet PLOS ONE’s publication criteria as it currently stands. Therefore, we invite you to submit a revised version of the manuscript that addresses the points raised during the review process.

We look forward to receiving your revised manuscript.

Kind regards,

Filipe Prazeres, MD, MSc, Ph.D.

Academic Editor

PLOS ONE

Journal Requirements:

Reviewers' comments:

Reviewer's Responses to Questions

**Comments to the Author**

1. Does the manuscript provide a valid rationale for the proposed study, with clearly identified and justified research questions?

Reviewer #1: Yes

Reviewer #2: Yes

2. Is the protocol technically sound and planned in a manner that will lead to a meaningful outcome and allow testing the stated hypotheses?

Reviewer #1: Yes

Reviewer #2: Yes

3. Is the methodology feasible and described in sufficient detail to allow the work to be replicable?

Reviewer #1: Yes

Reviewer #2: Yes

4. Have the authors described where all data underlying the findings will be made available when the study is complete?

Reviewer #1: Yes

Reviewer #2: Yes

5. Is the manuscript presented in an intelligible fashion and written in standard English?

Reviewer #1: Yes

Reviewer #2: Yes

6. Review Comments to the Author

You may also provide optional suggestions and comments to authors that they might find helpful in planning their study.

Reviewer #1: Thank you again for your work on the revised manuscript and the opportunity to review it. On the whole, you did a great job responding to the reviewer comments and I find the manuscript very much improved as a result.

I would like to offer one additional comment based on your revised text prompted by one of my original comments (#12). You have revised the sentence in question to read, "All data were anonymous and are not able to identify the participants' information." After reviewing this paragraph, I feel the key element to refine here has to do with the timing of the data use in question and subsequently the word tense to describe it.

"All data were anonymous" leads me to believe that this is an activity already taken and you were perhaps reviewing de-identified records as part of the screening process. But from the context, I believe you might be trying to convey thoughts about the security of the data you are assembling and using as part of this study. That is, you are maintaining the study data in a de-identified manner so any patient identifiers are not part of the analysis and the only way to know who a particular patient is would be to revisit a source document from a separate and secure location (locked file cabinet). If this is the case (referring to data security), the sentence could be made clearer. For example, "Data gathered are stored in the study database without identifiers and linking information is kept separate in a locked filing cabinet."

Thank you again for your attentive response.

Reviewer #2: Thank you for addressing my previous comments. In reading the revision, I only have 3 remaining comments:

1. Consider clarifying how participants' final status will be determined.

- There is a nice description of how reminders will be sent to participants at the time they are due for clinical follow-up. If the participant does not show up in clinic, how will the study team determine if they are deceased, transferred, or lost to follow-up?

- If none of these 3 dispositions are relevant to the given participant, will the follow-up be considered missed?

- When will the participant be considered a withdrawal? Is it if they actively state that they no longer wish to participate, or will the study team withdraw participants who continue to be no-shows to visits and/or unresponsive?

2. When will analyses be performed? Will analyses be based on a subset of the database population, i.e., a study population? If so, how will inclusion in the study population be determined, e.g., based on complete data, etc.? The RESTORE guidelines suggest that a flow chart as a possible way to present this information.

3. In the Informed Consent section, there are a few spots where the tense is confusing or mixed within sentence. You should consider revising this section to make it more clear that participants have already been consented and that the study team will continue to obtain informed consent, all per the ethical guidelines.

7. PLOS authors have the option to publish the peer review history of their article (what does this mean?). If published, this will include your full peer review and any attached files.

Reviewer #1: No

Reviewer #2: No

---

## [Author Response · Author response to Decision Letter 1]

4 Dec 2022

Response to Reviewers

PONE-D-21-27924R1 - [EMID:0dab44040bef7c5f]—entitled, “Clinical Outcomes, Patient-Reported Outcomes, and Economic Burden for Thai People Living with Chronic Urticaria (CORE-CU) in Routine Practice: A Study Protocol for a Monocentric Prospective Longitudinal Study”

Comments to the Author

Reviewer 1

Thank you again for your work on the revised manuscript and the opportunity to review it. On the whole, you did a great job responding to the reviewer comments and I find the manuscript very much improved as a result.

Kindly thanks you for your valuable suggestion. 

#1. I would like to offer one additional comment based on your revised text prompted by one of my original comments (#12). You have revised the sentence in question to read, "All data were anonymous and are not able to identify the participants' information." After reviewing this paragraph, I feel the key element to refine here has to do with the timing of the data use in question and subsequently the word tense to describe it.

"All data were anonymous" leads me to believe that this is an activity already taken and you were perhaps reviewing de-identified records as part of the screening process. But from the context, I believe you might be trying to convey thoughts about the security of the data you are assembling and using as part of this study. That is, you are maintaining the study data in a de-identified manner so any patient identifiers are not part of the analysis and the only way to know who a particular patient is would be to revisit a source document from a separate and secure location (locked file cabinet). If this is the case (referring to data security), the sentence could be made clearer. For example, "Data gathered are stored in the study database without identifiers and linking information is kept separate in a locked filing cabinet." Thank you again for your attentive response.

Thank you very much for your pertinent observation. We mage change the sentence per your recommendation accordingly under the “Informed consent, autonomy, and confidentiality” section as follows: 

“Data gathered are stored in the study database without identifiers, and linking information is kept separate in a locked filing cabinet. The unique identifiers will be generated, completely encrypted, coded, and will be recorded as double-entry for use, principally for statistical analysis purposes using computer software. Additionally, the collected data will remain confidential and shared only with authorized research members.”

Reviewer 2

Thank you for addressing my previous comments. In reading the revision, I only have 3 remaining comments:

#1. Consider clarifying how participants' final status will be determined.

- There is a nice description of how reminders will be sent to participants at the time they are due for clinical follow-up. If the participant does not show up in clinic, how will the study team determine if they are deceased, transferred, or lost to follow-up?

- If none of these 3 dispositions are relevant to the given participant, will the follow-up be considered missed?

- When will the participant be considered a withdrawal? Is it if they actively state that they no longer wish to participate, or will the study team withdraw participants who continue to be no-shows to visits and/or unresponsive?

Thank you very much for your concerns. Ultimately, participants will be followed and censored until the date of the earliest incidence of death, withdrawal for any reason, transfer to another center, or loss to follow-up, whichever occurred first. Based on our routine follow-up care, the communications team and case manager staff members will contact patients/caregivers based on participants’ preference contact methods (text message, phone, social media, or direct mail). In the case where there is no response, participants will be closed out and censored from the study as a loss to follow-up. 

Based on the Guideline for Good Clinical Practice, all participants have the right to withdraw informed consent at any time for any reason during the study period. In this case, data collected prior to study withdrawal or dropout due to any reason (based on the last observation carried forward) will be considered for participants who discontinue.

#2. When will analyses be performed? Will analyses be based on a subset of the database population, i.e., a study population? If so, how will inclusion in the study population be determined, e.g., based on complete data, etc.? The RESTORE guidelines suggest that a flow chart as a possible way to present this information.

Thank you very much for your comments. Preplanned statistical analysis will be performed based on one-year follow-up care. As mentioned, the last observation carried forward approach will be considered for data analysis. Regarding the whole sample population, multiple imputation methods will be used to analyze incomplete datasets for participants with missing covariates. However, variables containing more than 20% of missing data will be excluded from all analyses.

Moreover, for study reporting, we will disseminate the final results in line with the RECORD statement Guidelines (Item No.13)—an extension to the STROBE statement, which also includes the study flow diagram. 

Reference

- Benchimol EI, et al; RECORD Working Committee. The REporting of studies Conducted using Observational Routinely-collected health Data (RECORD) statement. PLoS Med. 2015;12(10):e1001885. doi: 10.1371/journal.pmed.1001885.

#3. In the Informed Consent section, there are a few spots where the tense is confusing or mixed within sentence. You should consider revising this section to make it more clear that participants have already been consented and that the study team will continue to obtain informed consent, all per the ethical guidelines.

Thank you for your suggestions. To make it more clear, we have to rewrite the “Informed Consent” section as recommended. 

Please read: “Written informed consent will be obtained from all participants before they are included in the study. The participants’ consent will be obtained by trained research assistants. Based on the International Conference on Harmonization: Thai Guideline for Good Clinical Practice, a thumbprint is obligatory when a participant is illiterate. All participants have the right to leave a specific question unanswered or withdraw informed consent at any time for any reason during the study period.”

---

## [Editor Report · Decision Letter 2]

12 Dec 2022

Clinical Outcomes, Patient-Reported Outcomes, and Economic Burden for Thai People Living with Chronic Urticaria (CORE-CU) in Routine Practice: A Study Protocol for a Monocentric Prospective Longitudinal Study

PONE-D-21-27924R2

Dear Dr. Chuamanochan,

We’re pleased to inform you that your manuscript has been judged scientifically suitable for publication and will be formally accepted for publication once it meets all outstanding technical requirements.

Kind regards,

Filipe Prazeres, MD, MSc, Ph.D.

Academic Editor

PLOS ONE
---

## [Editor Report · Acceptance letter]

10 Jan 2023

PONE-D-21-27924R2 

Clinical Outcomes, Patient-Reported Outcomes, and Economic Burden for Thai People Living with Chronic Urticaria (CORE-CU) in Routine Practice: A Study Protocol for a Monocentric Prospective Longitudinal Study 

Dear Dr. Chuamanochan:

I'm pleased to inform you that your manuscript has been deemed suitable for publication in PLOS ONE. Congratulations! Your manuscript is now with our production department. 

Kind regards, 

on behalf of

Prof. Filipe Prazeres 

Academic Editor

PLOS ONE